# Proper Dietary and Supplementation Patterns as a COVID-19 Protective Factor (Cross-Sectional Study-Silesia, Poland)

**DOI:** 10.3390/life12121976

**Published:** 2022-11-25

**Authors:** Karolina Krupa-Kotara, Mateusz Grajek, Anna Murzyn, Małgorzata Słoma-Krześlak, Karolina Sobczyk, Agnieszka Białek-Dratwa, Oskar Kowalski

**Affiliations:** 1Department of Epidemiology, Faculty of Health Sciences in Bytom, Medical University of Silesia in Katowice, 40-055 Katowice, Poland; 2Department of Public Health, Faculty of Health Sciences in Bytom, Medical University of Silesia in Katowice, 40-055 Katowice, Poland; 3Department of Basic Medical Sciences, Faculty of Health Sciences in Bytom, Medical University of Silesia in Katowice, 40-055 Katowice, Poland; 4Department of Dietetic, Faculty of Health Sciences in Bytom, Medical University of Silesia in Katowice, 40-055 Katowice, Poland; 5Department of Human Nutrition, Department of Dietetics, Faculty of Health Sciences in Bytom, Medical University of Silesia in Katowice, Jordana 19, 41-808 Zabrze, Poland

**Keywords:** COVID-19, SARS-CoV-2, immune system, dietary management, supplementation management

## Abstract

Background. The COVID-19 pandemic has become a challenge for the world today, so it is very important to create healthy eating habits in society to support immunity and raise awareness of the benefits of supplementation. Objective. The purpose of this study is to evaluate diet and dietary supplementation, since previous studies indicate the protective nature of these in building immunity during the pandemic and post-pandemic period (COVID-19). The hypothesis of the study is whether the dietary regimen presented by the COVID-19 respondents can be considered protective in building immunity against SARS-CoV-2. Material and methods. The study included 304 subjects, with an average age of 39.04 ± 23.59. The main criteria for inclusion in the study were that the respondent was ≥18 years old and participated voluntarily. The study was conducted using an original questionnaire. Results. In the study group, no change was noticed in the previous diet during the COVID-19 pandemic, while the level of physical activity among the respondents decreased. Too low a percentage of people regularly consumed, among other foods. legume seeds—only 10.5% of respondents consumed them several times a week—and citrus fruits—the largest number of respondents, as many as 39.8%, only consumed them several times a month. The largest percentage of respondents with mild/scanty COVID-19 disease regularly took vitamin C-containing preparations (*n* = 61; 59.80%). Statistical analysis showed that there was a correlation between the incidence of mild/scanty COVID-19 and the regularity of taking vitamin C-containing preparations (T = 11.374; r = 0.611; *p* = 0.04603). A statistical significance level was also obtained for the regularity of supplementation of multivitamin preparations, which were taken by 68% (34) of respondents affected by mild/scanty COVID-19 (T = 13.456; r = 0.711; *p* = 0.02191). Conclusions. The study’s hypothesis was supported. Respondents characterized by a normal dietary pattern and taking supplements commonly recognized as immune “boosters” were more likely to mildly survive COVID-19. Moreover, it was shown that the pandemic in most of the respondents did not significantly affect their dietary strategy. It is reasonable to conclude that the dietary patterns adopted may be a common way to prevent SARS-CoV-2 infections and their possible complications.

## 1. Introduction

The coronavirus disease (COVID-19) pandemic caused by the severe acute respiratory syndrome coronavirus (SARS-CoV-2) has affected many spheres of human life, leading to changes in virtually every aspect [1]. Since then, the scientific world has produced a plethora of important scientific studies that have addressed the characteristics of the COVID-19 virus and the mortality caused by it. An aspect of great importance in the era of the pandemic is immunity, which, through appropriate action, can prevent COVID-19 incidence and mitigate the course of the disease, thereby reducing the number of deaths caused by the coronavirus [1,2].

The immune system, to work properly, must perform four basic tasks. The first is to prevent microorganisms from entering the human body. If, despite the barrier created, microorganisms do enter, the immune system is obliged to recognize them and determine whether they pose a threat to the host body. The third task of the immune system is to fight against agents that have previously been identified as harmful and dangerous, by activating several cell types and a series of immune processes in the body. The last task of the immune system is to create immune memory after an encounter with a pathogen. Thanks to this ability, the next encounter with a harmful microorganism will be more benign for the body, as the response is much faster and stronger [2,3].

Such efficient functioning of the immune system is possible due to the formation of several types of cells, which are responsible for many different tasks in ensuring an adequate immune response after a harmful agent enters the body. The immune system is tasked with producing the body’s aforementioned immunity, which can be divided into acquired and innate immunity, which complement each other to form a functional whole [4]. Innate immunity is the first line of defense against microorganisms in all higher organisms from the moment of birth and produced immediately after the first contact with a pathogen, but its action is not specified, as in the case of acquired immunity. The first barrier to microorganisms in acquired immunity can be found in the form of a skin barrier, mucous membranes, or special substances secreted by the body, e.g., gastric juice, characterized by a low pH or bacterial flora in the intestines, i.e., the so-called passive mechanisms of innate immunity. When the first line of defense fails, active mechanisms in the form of special cells with phagocytic properties (e.g., macrophages or monocytes) become involved [5,6,7]. Acquired immunity is characterized by action through T and B lymphocytes and the synthesis of many antibodies and receptors. This immunity can develop even days after the first contact with a microorganism and is more precise than innate immunity, as it targets one specific pathogen. One can speak of naturally acquired immunity when antibodies are produced after natural contact with the microorganism and a history of infection, while artificially acquired immunity can be acquired, for example, by receiving a vaccine. In this way, antibodies are produced in the body against a specific pathogen, but without the obvious symptoms of disease caused by it [8,9].

There are many factors that more or less influence the corresponding immune response. Non-modifiable factors over which society has little influence include genetics, gender, time of day, or the stage of life a person is in (such as childhood or old age). Issues that an individual can influence are modifiable factors that should be taken care of, thus supporting the immune system and its response to microorganisms. These factors can include an individual’s susceptibility to stress, the onset of obesity, lifestyle (inadequate sleep hygiene, lack of physical activity, smoking, alcohol consumption), or inadequate diet [10,11,12].

Currently, there are no effective treatments to adequately treat people infected with SARS-CoV-2, so it is very important to properly support the immune system, which, in addition to immunization, is an effective method of prevention of COVID-19. A very important point to mention in terms of building immunity is a well-balanced diet, which can provide the body with all the necessary nutrients, elements and vitamins. Particularly noteworthy are vitamins A, D, E and C which show special properties in supporting the body’s immunity, as well as in the milder course of illness. Similar properties are characterized by elements such as zinc and selenium. When composing an appropriate diet, special attention should be paid to an adequate supply of probiotic substances and prebiotics, which, influencing the proper functioning of the intestines, strengthen the body’s immunity. Inherent in a properly composed diet is an adequate amount of daily physical activity, sleep and reduced exposure to stress [13,14,15].

A very important factor in the proper functioning of the human immune system is a properly composed diet. This is the basis for ensuring the maintenance of human health. Due to a lack of time and an excess of daily duties, very often people’s daily diet is rich in processed foods, containing large amounts of sugars and saturated fats, food additives and preservatives. Such a diet is at the same time poor in vitamins, fiber and substances that exhibit antioxidant properties. The described dietary pattern leads to the consumption of too many calories and to gradually appearing nutritional deficiencies, which significantly impair the proper functioning of the immune system and increase the risk of infections and diseases of civilization, such as obesity. Deficiencies of important components, such as vitamin A, vitamin C and selenium, among others, lead to increased susceptibility of the body to infections. An action to prevent such a phenomenon is nutritional intervention, which can be provided, for example, by a specialized nutritionist, who, through a properly collected nutritional history and clinical examination, can assess potential nutritional deficiencies that negatively affect the body. Thanks to such a diagnosis, it is possible to determine the goal of diet therapy, establish the scheme according to which it will be implemented and put into practice an appropriately selected regimen of action under the guidance of a specialist who can properly compose a nutritional pattern to compensate for existing deficiencies and prevent further deficiencies. A rational diet is composed in such a way that it provides the right amount of protein, fats and carbohydrates and is rich in all vitamins, elements and dietary fiber, which ensures, among other things, the proper functioning of the intestines. In addition to a properly balanced diet is the inclusion in the daily menu of products and substances that support and stimulate the immune system. Such products include, for example, fish oils, which are rich in fatty acids, especially those of the omega-3 family (eicosapentaenoic acid–EPA, docosahexaenoic acid–DHA), garlic, or onions, which are natural antibiotics that strengthen the immune system. Herbs and plants with appropriate immunomodulatory properties, such as aloe vera, purgative, or raspberry fruit, which exhibit anti-inflammatory and antipyretic effects, among others, also play a special role in supporting human immunity. The implemented nutritional therapy should be constantly monitored and its effects controlled by a specialist. Such control can be ensured by regular meetings with the patient, at which a nutritional history will be taken to determine the patient’s current nutrition and nutritional education will be provided to avoid further mistakes in the future, mistakes which can contribute to deficiencies, thereby weakening the immune system. In the case of deep, persistent deficiencies that cannot be compensated for by diet alone, it is worth considering supplementation with preparations rich in appropriately selected substances, such as vitamin A, D, selenium, or multivitamin preparations [16,17,18,19,20].

Therefore, the purpose of this study is to evaluate diet and dietary supplementation, since previous studies indicate the protective nature of these in building immunity during the pandemic and post-pandemic period (COVID-19). The hypothesis of the study is that the dietary regimen presented by the COVID-19 respondents can be considered protective in building immunity against SARS-CoV-2.

## 2. Materials and Methods

### 2.1. Study Area

The survey was conducted using the Computer-Assisted Web Interview (CAWI) method, by sharing the questionnaire on online forums and social media. Data was collected from March 2022 to May 2022.

Because the survey was conducted by questionnaire using the CAWI method (a web-based form), the authors used several safeguards to ensure that the bias that often affects surveys of this type was minimized. Respondents received links to the questionnaire in their mailboxes or via closed groups in online forums. In addition, to avoid the phenomenon of fake/bot responders, access keys and CAPTCHA (Completely Automated Public Turing test to tell Computers and Humans Apart) codes were introduced to verify that a person was not an automaton filling out the surveys. In addition, survey completion times and login periods were checked. The estimated time for filling out surveys was 4–7 min, and time below these values was excluded. In addition, questionnaires that indicated biased filling, always choosing the same option and skipping open-ended questions (requiring the entry of a value or a short verbal answer) were rejected. Thus, the questionnaire’s return rate was estimated at 84%.

### 2.2. Eligibility Criteria and Study Sample

The main criterion for inclusion in the study was that the respondent was at least 18 years old, declared residence in Silesia (Poland) and had a history of SARS-CoV-2 infection. The questionnaire was filled out correctly 304 times. The fact that respondents had contracted COVID-19 was self-reported. Respondents were asked about the fact of having contracted COVID-19, the date of infection, its severity (severity of infection was assessed subjectively) and the type of testing method (self-test, point-of-care test, etc.).

The necessary sample size was calculated, depending on the size of the population of the Silesian region, Poland. It was estimated that a sample of 304 students would be sufficient and representative of the Silesian region. It was assumed, according to the CSO (Central Statistical Office) Report, that the general population is 4615.9 thousand people. The sample size was calculated according to the formula:
Nmin = NP ⋅ (α^2^ ⋅ f(1 − f)) ÷ NP ⋅ e^2^ + α^2^ ⋅ f(1 − f)
where

Nmin-minimum sample size;NP-the size of the population from which the sample is drawn;α-confidence level for the results;f-the size of the fraction; e-assumed maximum error.

For the population of Silesia (Poland), the minimum sample size of respondents was calculated, which was 138 people (α = 0.95; f = 0.9; e = 0.05). Based on these calculations, the collected group was considered representative. 

### 2.3. Ethical Approval

The study design, in light of the Act of December 5, 1996, on the professions of physician and dentist (Journal of Laws of 2011, No. 277. item 1634, as amended), is not a medical experiment and does not require evaluation by the Bioethics Committee of the Silesian Medical University in Katowice, as it is based on the patient’s own experience. In addition, the data collected was based on an anonymous questionnaire, to which patients gave written, voluntary consent; the absence of a completed questionnaire also meant that patients did not consent to participate in the study. The study was conducted by the ethical principles of research based on the Declaration of Helsinki, as amended.

### 2.4. Research Tool

The survey was conducted using an author’s questionnaire containing 25 questions. The questionnaire was divided into two parts. The first contained questions related to gender, year of birth, weight, height, place of residence, education and type of work. The second part consisted of questions related to health status, physical activity, eating habits, frequency of consumption of certain products and supplementation used. The survey consisted mostly of closed-ended single- or multiple-choice questions. The questionnaire was validated and a sample survey was conducted on a group of 30 people. The questionnaire for the pilot group additionally included the opportunity to impress opinions on the questions and to add suggestions and possible changes to the content. All suggestions were applied and the survey was conducted again with a group of 30 people who had already used the final version of the questionnaire. This procedure resulted in a Cohen’s kappa coefficient value of κ = 0.82, indicating very good agreement.

### 2.5. Analysis

Diet quality was assessed using the Diet Quality Index (DQI) reminders. Each dietary characteristic (food group) was scored. The general interpretation of the results assumed a maximum score of up to 16, but a score closer to 0 indicated a better quality of the dietary pattern maintained.

The results were statistically analyzed using Statistica 13.0 software, using the chi-square and Kruskal-Wallis test to detect differences between groups. The results obtained were considered statistically significant when *p* < 0.05.

## 3. Results

The study included 304 participants (39 ± 19), including 194 women and 110 men between the ages of 18 and 81 (Table 1). 

Respondents between the ages of 18 and 35 made up the largest group among those surveyed 189 (62%). The second most numerous group was those in the age range between 36 and 55, who accounted for 64 (21%) of the total. The smallest percentage of respondents were those between the ages of 56 and 81, 51 (17%). The assessment was also made taking into account the place of residence. The largest number of respondents, i.e., 95 (31%), declared their place of residence as a city with a population of more than 150,000. A large group of respondents also lived in rural areas: 84 (28%) respondents. The least number of people, only 51 (17%) respondents, declared that they reside in a city with a population of up to 50,000. Based on the assessment taking education into account, it was noted that more than half of the respondents, as many as 164 (54%), declared a higher education. Slightly fewer, 96 (32%) of the respondents, had only secondary education. The smallest percentage of people, only 32 (11%) was a group of respondents declaring vocational education and 12 (4%) only primary education. Among those surveyed, the largest group, as many as 117 (38%) of respondents, declared that they had white-collar jobs. On a similar level were those doing manual labor, 68 (22%) of respondents, and those with the status of pupil/student, 65 (21%) of all respondents. Retirees/retirees made up a group of 45 (15%), while only 9 (3%) of those surveyed declared themselves as not working. Among the active respondents, more than half, 176 (59%), declared their occupation as non-medical. Only 50 (16%) of all those surveyed work in the medical sector.

Nearly half (51.6%) of respondents had been diagnosed with COVID-19, while 27% of respondents had not received such an official diagnosis, but had self-tested. Some respondents (21.4%) were unsure whether they had undergone SARS-CoV-2 infection, as symptoms were uncharacteristic or very mild. The vast majority (74%) of respondents said they had undergone COVID-19 vaccination. Respondents rated their health as good (38%) and rather good (34%). The highest rating (very good) for health condition was given by 21% of respondents, while only 6% of respondents rated their health condition as bad. More than half (62.5%) of the survey participants were diagnosed with chronic non-communicable diseases (NCDs), such as hypertension (17.4%), endocrine disorders (9.9%), type I/II diabetes (7.9%), atherosclerosis (6.9%) and ischemic heart disease (5.3%). The smallest percentage of respondents reported having respiratory diseases (2.6%), cancer (1.3%) and dyslipidemia (1.3%). The remaining survey participants (6.6%) were diagnosed with other non-communicable diseases. Half of the respondents reported no symptoms of infection in the last six months preceding the survey and 36% had experienced infection twice. Seven percent of respondents were sick once a month. The smallest percentage of respondents had gone through an infection several times in 6 months (5%) and several times a month (2%). Throat and ear infections, as well as diarrhea, were the most common symptoms among those with infectious diseases. 

The level of physical activity during the COVID-19 pandemic in 44% of respondents remained unchanged from the pre-pandemic state, while physical activity decreased in a similar number (43%) of respondents. An increase in the level of physical activity was declared by only 13% of survey participants. More than half of the respondents (51%) rated their level of physical activity as low (sedentary lifestyle, daily household chores, walking 1 h/week). Activity at a moderate level (e.g., trotting, cycling at least 2 times a week) was reported by 36.5% of the surveyed individuals. Only 12.5% of respondents rated their physical activity as high (regular high-frequency physical activity at least 4–5 times per week).

Stress caused by the COVID-19 pandemic was moderately felt by more than half (57%) of respondents, while high and very high levels of stress affected 21% and 4% of respondents, respectively. No feeling of stress was declared by 18% of those surveyed.

A sleep duration of 5–7 h/day applied to the majority of respondents (62%). A smaller percentage of survey participants (32%) reported a longer sleep duration of 7–9 h/day, while only 2% of respondents sleep more than 9 h/day. Shorter sleep duration (less than 5 h/day) concerned 4% of respondents. In the vast majority of respondents (74%), sleep duration did not change during the COVID-19 pandemic.

The duration of the COVID-19 pandemic brought no change in diet in 47.7% of respondents. An increase in food consumption compared to the time before the pandemic was observed in 26% of respondents and the opposite relationship was observed in 8.9% of respondents. In opposition to 7.2%, 16.4% of respondents pay more attention to the type of products they eat and 12.5% eat more healthily. Deterioration in their diet was declared by 10.5% of respondents.

The regularity of taking preparations containing specific substances is shown in Table 2. 

The vast majority, 73.7%, of respondents do not take vitamin A, or preparations rich in it; regular intake is declared by 15.1% of respondents and irregularly vitamin A is taken by 34 (11.2%) respondents. As for vitamin C, 41.1% of respondents do not take it at all, 102 (33.6%) declare regular supplementation and 25.3% irregular. Regular supplementation with vitamin D is declared by 46.1% of respondents. Irregular consumption is declared by 26% of people and 28% of respondents do not supplement vitamin D. In addition, 76.3% of the respondents do not supplement vitamin E and regular and irregular intake is declared by the same number, 11.8%, of the respondents. Analyzing the respondents’ intake of multivitamin preparations, most, 63.5%, do not take them at all, irregular supplementation is declared by 20.1%, while regular supplementation is declared by only 16.4% of the respondents. Regular selenium supplementation is declared by only 8.2% of respondents, while at the same time 63.5% do not take it at all. In the case of zinc, the vast majority, as many as 74.3%, declare no supplementation, with only 15.1% of respondents taking it regularly. Regular intake of probiotics is declared by only 4.9% of those surveyed, while the vast majority at 80.3% do not take probiotics at all. Fish oils or oil rich in omega-3 fatty acids are consumed regularly by only 22.7% of the subjects and irregularly by 23%. Lack of supplementation with products rich in omega-3 acids is declared by more than half of the 54.3% of those surveyed. 

The educational level of respondents had a statistically significant (T = 12.387; r = 0,743; *p* = 0.04272) effect on the regularity of taking multivitamin preparations. Lack of supplementation was the most frequently given answer in all groups, while those with primary education (83.3%) and vocational education (75%) reported this lack the most. 

Table 3 shows the incidence of COVID-19 disease according to regular intake of preparations containing certain substances.

The largest percentage of respondents with mild/scanty COVID-19 disease regularly took vitamin C-containing preparations (*n* = 61; 59.80%). Statistical analysis showed that there was a correlation between the incidence of mild/scanty COVID-19 and the regularity of taking vitamin C-containing preparations (T = 11.374; r = 0.611; *p* = 0.04603). A statistical significance level was also obtained for the regularity of supplementation of multivitamin preparations, which were taken by 68% (34) of respondents affected by mild/scanty COVID-19 (T = 13.456; r = 0.711; *p* = 0.02191).

The frequency of consumption of specific products by respondents is shown in Table 4. The consumption of red meat and poultry several times a week is declared by 45.7% of respondents, while 7.2% consume them occasionally. 47% of respondents do not consume offal at all, while only 1.6% declare consumption several times a week. Half of the respondents consume fish several times a week, while 32.2% do not consume fish at all. The vast majority, as many as 60.5% of respondents, do not consume seafood, while 9.2% consume it several times a month. More than half of respondents (56.9%) say they consume eggs several times a week, while 5.9% do not consume them at all. Milk and dairy products are consumed several times a week by 39.5% of respondents, while 3% do not consume milk and dairy products at all. 27.3% of respondents consume fermented dairy products several times a week, while 21.7% do not consume them at all. Consumption of rennet-ripened cheeses several times a week is declared by 38.8% of respondents, while 4.9% do not consume these products at all. 35.5% of respondents consume cottage cheese several times a week, 107 (35.2%) consume it several times a month and 5.6% do not consume it at all. Whole-grain products are consumed several times a week by 36.5% of respondents, 21.4% consume them once a day, while 6.6% of people consume whole-grain products only occasionally. Pickles are consumed by 49% of people several times a month, while 4.9% do not consume pickles at all. Citrus fruits are consumed several times a month by 39.8% of respondents, while 2.3% do not consume citrus at all. Garlic and onions are consumed several times a week by 51.3% of respondents, once a day by 12.8% of respondents, and 6.3% do not consume these products at all. 34.2% of respondents consume nuts occasionally, 26% consume them several times a month and daily consumption is declared by only 3.6% of respondents. The largest percentage, as many as 35.2% declare that they consume seeds occasionally, 48.8% do not consume them at all, while 2.6% consume them once a day. Legume seeds are consumed several times a month by 35.9% of respondents, while 9.5% do not consume them at all. Vegetable fats are consumed several times a week by 39.8% of respondents, daily consumption is declared by 17.8% of people, while 3.3% do not consume them at all. 28% of people declare consuming vegetable fats several times a week, 18.4% consuming them daily and 9.5% do not consume animal fats at all. 28% of respondents consume herbs several times a week, 15.5% consume them several times a day and 9.2% do not consume them at all.

Based on an analysis of the consumption of each food group, the respondents’ dietary pattern was evaluated by calculating the DQI. It was found that respondents showing low DQI values (0–6 points) were more likely (83.2%) to have experienced mild or scanty COVID-19. Respondents who scored higher values in the assessment (>7 points) underwent COVID-19 in a worse self-assessed form (71.3% of this group). This relationship was confirmed statistically (T = 15.020; r = 0.698; *p* = 0.23450).

## 4. Discussion

In the current global situation, the body’s immunity is an aspect that should be specially taken care of. A properly balanced diet, rich in all macronutrients and micronutrients, affects the functioning of the immune system positively. In a situation where the daily diet is unable to provide adequate amounts of essential nutrients and vitamins, it is important to consider implementing supplementation. Immunity is defined as the totality of defense mechanisms that protect against disease-causing pathogens in the body [21]. Nutritional status plays a key role in the immunomodulatory effect in the course of viral infections. The process of phagocytosis, activation of cytokines and reduction of oxidative stress by fighting free radical molecules contribute to the health-promoting spectrum of the immune system. This process can be stimulated by biologically active compounds present in nutrients. Enriched foods or a balanced diet including vitamins and minerals C, A, E, zinc, and especially increased daily intake of cholecalciferol-vitamin D in supplement form, contribute to fighting and preventing the proliferation of SARS-CoV-2 virus cell particles and minimizing mortality and complications resulting from COVID-19 disease [22]. The relationships described in the study are important in that they also have their role in post-COVID prophylaxis and affect the building of lifelong immune reserves protecting against repeat infections and the development of a dangerous course of infection [23]. Along with proper nutrition, the functioning of the body is also influenced by regular daily physical activity, which mobilizes the body for action, and an adequate amount of time per day for sleep and rest [24].

The study assessed dietary habits among the respondents which can positively or negatively affect the human immune system. In the case of pork, mutton beef and poultry consumption, the results were satisfactory, as up to 45.7% of respondents consume these meats several times a week, which is positive due to the content of zinc, iron and B vitamins, among other elements, in these products. Similar results were obtained in the study by Moskal and Michalska [25], who showed that more than half (55.2%) of the respondents eat meat several times a week. Different results were published in the work by Piejko et al. [26], where the vast majority of respondents, more than 70%, consumed meat several times a week. Offal, and in particular poultry liver, show many health benefits as a rich source of iron and vitamins A and B, which support immunity. Among those surveyed, the consumption of offal ranks low, as 36.8% of respondents consume it occasionally, while as many as 47% do not consume offal at all.

Nutrients that support the work of the human body include unsaturated fatty acids, especially those of the omega-3 family, vitamins A, D, E, and selenium. They can be found in very high amounts in fish, for example. The survey showed that half of the respondents (50%) eat fish several times a month. In the case of seafood, which is rich in selenium and zinc, consumption ranks very low, as more than half of the respondents (60.5%) do not consume these products at all. Slightly different results were obtained in the study by Stoś et al. [27], where fish consumption was at a very low level, with the majority of respondents consuming these products 1–3 times per month, or less frequently. Lower fish consumption was reported in the women’s group.

Eggs are commonly referred to as a treasure trove of vitamins and minerals, as they are a rich source of vitamin D, B vitamins, selenium, zinc, iron, etc. Thus, regular consumption of this product provides the human body with many substances that support the immune system [28]. Our study showed that more than half of the respondents (56.9%) consume eggs several times a week, which is in line with the results obtained in the study conducted by Cegielska-Radziejewska et al. [29], where consumption of eggs 2–3 times a week was declared by the largest group of respondents.

Speaking of products rich in immune-supporting components, we should also mention milk and dairy products, which are rich in fat-soluble vitamins, B vitamins and elements such as zinc and iron. Fermented dairy products are also characterized by special properties, which, in addition to vitamins and minerals, are also rich in probiotics, which, having a positive effect on intestinal function, contribute to more effective working of the immune system [30]. The results obtained in our study allow us to conclude that the largest percentage of respondents (39.5%) declare that they consume milk and dairy products several times a week, which ranks far below the recommendations recommended by the IZŻ (Institute of Food and Nutrition). In the case of consumption of fermented dairy products, the situation is better, with the largest number of respondents (27.3%) declaring consumption of these foods several times a week. Similar results were obtained by Malczyk et al. [31] and Grębowiec and Korytkowska [32], where the largest percentage of respondents declared consumption of milk and dairy drinks several times a week, and Misiarz et al. [33], who in their study showed that the recommended portion of milk and dairy products was consumed by only 10% of respondents. The same study also found underconsumption of fermented dairy products, where the daily recommended serving of this product was consumed by only 15% of respondents.

According to the recommendations of the Institute of Food and Nutrition, whole-grain products such as bread, rice, groats and pasta should form the basis of daily nutrition, as they are a source of many vitamins and elements that support the human body. Our study showed that these products are consumed several times a day by only 19.7% of respondents, with the largest percentage of people (36.5%) declaring that they consume whole grains several times a week. Similar results were obtained in the study by Stoś et al. [27], where consumption of whole-grain products a few times a week was most common, with daily consumption of these products declared by the smallest percentage of subjects. The results obtained in our study can also be equated with the results of the work of Misiarz et al. [33], where the intake of whole-grain products was also at an insufficient level, but the more frequent consumption of this group of products were characterized by people associated with medical faculties.

From a nutritional point of view, pulses belong to a group of products with very high health benefits. Due to their high content of fiber, minerals and vitamins, they affect the proper functioning of the human body, which is why it is recommended to consume them at least twice a week [34]. In the study conducted, the results obtained are not satisfactory, as 41.8% of the surveyed people declare only occasional consumption of legumes and only 10.5% of respondents consume this product as recommended several times a week. Similar results were published in the work by Stoś et al. [27] where the largest percentage of respondents declared consuming legumes 1–3 times a month or less frequently. Such a low consumption of this food product in the study group may be related to a lack of knowledge about the health benefits associated with its consumption and insufficient knowledge of how to cook legumes.

The situation caused by the COVID-19 pandemic has changed the previous lives of everyone in the world, affecting health, lifestyle and economic situation. Our study addressed the issue related to changing the current diet and eating habits among the respondents. The results obtained make it possible to determine that in almost half (47.7%) of the respondents, the COVID-19 pandemic has not affected the way they eat so far, with 26% of the respondents declaring an increase in the food they eat. Similar results were obtained by Olearczyk and Walewska-Zielecka [35], who in their work showed that, for 60% of respondents, the pandemic situation did not affect dietary change, and by Rzadkowolska [36], who in her work noted an increase in the amount of food previously consumed by respondents, as well as paying less attention to the type of meals consumed and increasing the proportion of snacks, fast food and salty snacks in the usual diet.

If there is an insufficient supply of certain substances in the diet, consideration should be given to taking them through regular supplementation. In our work, respondents were asked questions about their use of supplementation with selected drugs, preparations, or supplements. The results obtained were not satisfactory. In the case of vitamin A, E, selenium, zinc, multivitamin preparations and probiotics, the vast majority of respondents declared that they do not use supplementation with selected preparations or substances. Only in the case of vitamin C, D and omega-3 fatty acids were different results obtained. In the case of cholecalciferol, 46.1% of respondents declared regular supplementation of this vitamin, while 26% declared irregular supplementation. Considering ascorbic acid, regular and irregular supplementation was declared by 58.9% of respondents. Omega-3 fatty acids in the form of cod liver oil and fish oils are taken by 45.7% of those surveyed. The results obtained in this study were similar to the results of the study taking into account a group of vegetarians and people eating a traditional diet. Supplementation with specific preparations or substances ranked low; however, a greater willingness to supplement was observed among the group of vegetarians. The question used a chi-square test, which took into account the respondents’ education in its analysis. The test used showed a significance between education and respondents’ use of multivitamin preparations. In all other cases, the chi-square test proved statistically insignificant, so it can be concluded that education did not affect respondents’ intake of specific supplements. The question on the use of selected supplements was also analyzed taking into account the incidence of the respondents’ COVID-19 disease. The chi-square test was also used to conduct this analysis. This analysis showed a significant relationship between the incidence of COVID-19 disease and the intake of multivitamins and vitamin C preparations. In all other cases, no relationship was shown.

Several scientific studies [37,38,39,40] have shown that a properly composed diet can significantly affect the body’s immune system and thus have a protective effect against SARS-CoV-2 infection [41]. Of particular note is the relationship between vitamin D levels and the number of infections, the severity of illness and mortality from COVID-19 [37]. Vitamin D has pleiotropic effects; receptors for this vitamin have been found in almost all cells, including those of the immune system and 25 OH-D stimulates cytokines, accelerates the immune response and has anti-inflammatory effects, which may be of particular importance in SARS-CoV-2 infection [42]. A cohort study was conducted in Switzerland to evaluate the relationship between SARS-CoV-2 test results and plasma vitamin D levels. The presence of SARS-CoV-2 was tested by PCR using nasopharyngeal swabs. Analysis of the results showed that positive COVID-19 cases had a significantly lower median 25OH-D than negative COVID-19 cases. Some researchers have also suggested that vitamin D levels affect the severity of COVID-19 [43]. Observations in selected European countries confirm that there is a negative relationship between mean serum 25OH-D levels and the number of COVID-19 cases per million population; the same relationship exists between vitamin D levels and the course of the disease and the number of COVID-19 deaths [44]. However, the effect of 25OH-D on the frequency and severity of infection with the new type of coronavirus remains unclear and requires additional research. According to current recommendations, however, vitamin D supplementation is recommended [45]. In addition, some studies [46,47] have shown that vitamin D, in combination with zinc and vitamin C, is an integral part of the immune system and exhibits synergistic functions at various stages of defense, such as in maintaining the integrity of biological barriers and the functionality of cells that make up the immune system. Therefore, a deficiency of these key components can lead to damage to mucosal epithelial cells and perhaps make them more susceptible to penetration by pathogens such as SARS-CoV-2.

In a study by Zhou et al. [48], some affinity between body calcium levels and the course of SARS-CoV-2 infection was pointed out. In the clinical study, mild and moderate cases showed unsatisfactory calcium levels in the early stages of viral infection, while severe and critical cases showed significantly lower calcium levels than mild and moderate cases in the early stages. It was also found that low calcium levels were associated with multi-organ damage in the course of COVID-19. The antioxidant potential of many vitamins is also important and many current scientific reports indicate that antioxidants in the diet and dietary supplements can significantly affect the course of COVID-19. In their study, Biancatelli et al. [49] demonstrated that the simultaneous administration of quercetin and ascorbic acid is an experimental strategy for the prevention and treatment of infections with certain respiratory viruses (such as SARS-CoV-2). Blocking viral penetration is a key strategy and quercetin impedes viral membrane fusion in both influenza [50] and SARS-CoV and MERS-CoV in vitro studies [51]. Other studies [52,53,54,55] have shown that many polyphenols can be an inexpensive and safe prophylaxis to reduce viral infectivity and cytokine storm risk. At the molecular level, polyphenols act as inhibitors of viral proteases that participate in viral replication due to their general affinity for proteins through hydrogen bonding and low risk of toxic effects. The same may be true of the binding of these compounds to protein S (the virus spike). Recommendations for the general population in the post-pandemic period should focus on health-promoting dietary patterns with possible supplementation to compensate for nutritional deficiencies [56]. These can be generally described as rich in plant-based foods, including fresh fruits and vegetables, soy and nuts, which are good sources of antioxidants [57] and omega-3 fatty acids and low in saturated and trans fats, animal protein and added/refined sugars [58]. Most of these dietary goals can be achieved with the traditional Mediterranean diet [59], which is rich in protective and anti-inflammatory polyphenols [60]. The need for dietary and lifestyle changes in the post-pandemic period depends on physiological, sociological, economic and habitual factors. Malnutrition can occur even when a person has access to an adequate amount of food with a varied assortment. Several studies indicate the association of both quantitative and qualitative malnutrition with the development of many infectious diseases [48,49]. Malnutrition consists of an inadequate supply of major nutrients, such as (protein, fat and carbohydrates), but also minerals and vitamins, which have been widely described above. Therefore, it is necessary to identify solutions that would give rise to model strategies for strengthening immunity. In addition to country-specific standards, various nutrition and nutritional programming models are useful, such as Linear Programming proposed by Corné van Dooren [61] and the Emerging Programming Model proposed by Benvenuti and De Santis.

According to a definition published by the WHO (World Health Organization), health is defined as, “the total physical, mental and social well-being of a person and not merely the absence of disease or infirmity. A person’s health is influenced by many aspects that include lifestyle (diet, physical activity, ability to cope with stress and frequency of use of stimulants), physical environment, genetic load and access to health care. In our study, respondents were asked to subjectively assess their health status during the COVID-19 pandemic. The results obtained can be considered satisfactory, as most people define their health status during the COVID-19 pandemic as good (38%) and rather good (34%). Similar results were obtained by Wicka [45] in her study, who found that the majority (71.4%) of respondents did not experience any deterioration in their health during the COVID-19 pandemic and after the SARS-CoV-2 infection. The good health during the COVID-19 pandemic declared by the respondents may be related to the age of the respondents, as the majority of the respondents were young, enjoying full health and the absence of comorbidities.

Physical activity along with a healthy, well-balanced diet significantly affects the functioning of the entire body. Regular activity reduces high levels of stress, improves mental health and strengthens a person’s strength and motor skills. Moreover, it supports and strengthens the functioning of the immune system and prevents the onset of diseases such as hypertension, cardiovascular disease, type II diabetes and cancer [62,63]. According to WHO recommendations, adults between the ages of 18 and 64 are advised to perform moderate-intensity physical activity for 150–300 min per week, or high-intensity physical activity for 75–150 min/per week. For the elderly over 65 or those with comorbidities, it is recommended to undertake physical activity after prior consultation with a physician [64]. In our study, the respondents were asked to evaluate their physical activity and the change in its level during the COVID-19 pandemic. The results obtained in the study were not satisfactory because more than half (51%) of the respondents rated their level of physical activity as low and 43% of the respondents admitted that during the pandemic their previous physical activity had decreased. Taking into account the recommendations published by the World Health Organization, it can be concluded that the level of physical activity among those surveyed is too low, as most of them lead sedentary lifestyles, performing only daily chores. Mucha and Mucha [65] were also asked in their study to assess the change in physical activity during the coronavirus pandemic. Her results were very similar to those obtained in her work, as the vast majority of respondents (75%) declared a decrease in physical activity levels during the pandemic era. The reason for the low level of physical activity among respondents may be the fact that during the pandemic, all sports facilities, including gyms, fitness clubs and swimming pools, remained closed for long periods. In addition, it is very hard, sometimes impossible, for most people to motivate themselves to undertake physical activity at home (exercise using the Internet) or outdoors.

Stress is the habitual reaction of the human body to all things, phenomena, or events that are uncomfortable and cause a state of mental tension. The COVID-19 pandemic changed the lives of people, putting them in a situation of danger and uncertainty about the future. Sustained high levels of stress over a long period negatively affect the human body by, among other factors, weakening the immune system and increasing susceptibility to the development of diseases and infections [66]. Given the exceptional situation in the world, respondents were asked about the level of stress they felt during the duration of the COVID-19 pandemic. The results were surprising, as more than half of the respondents (57%) declared that the level of stress caused by the COVID-19 pandemic remained at a moderate level. A different situation was presented in her work by Dymecka [67], who emphasizes that the stress caused by the COVID-19 pandemic in people in different countries around the world remained at a high level.

A very important aspect of a person’s daily life is the appropriate amount of sleep per day. The most effective performance of the body is observed in a person who gets 7 to 9 h of sleep per day. In case of insufficient sleep, reduced concentration, apathy, irritability and reduced mobilization of the body to fight pathogens of disease can occur. Lack of adequate rest can also lead to serious metabolic disorders, resulting in type II diabetes or obesity [68]. The COVID-19 pandemic was a tremendous stressor for many people, which could have significantly altered the amount of sleep they had and impaired its quality [50,69,70]. Our study addressed the issue of sleep and the change in its previous quantity caused by the pandemic. The results show that the largest percentage of people (62%) devote 5–7 h per day to sleep, which is too little. When asked about the change in the amount of previous sleep caused by the COVID-19 pandemic, the vast majority (74%) of respondents declared that there was none, which is in line with the results obtained by Olearczyk and Walewska-Zielecka [35], who showed that 67% of respondents during the pandemic period did not notice a change in the amount of sleep.

In conclusion, a well-balanced diet, rich in vitamins and minerals, enriched with rational supplementation and regular physical activity, can ensure the human body’s proper functioning and the proper work of the immune system. Therefore, it is worth regularly raising people’s awareness of the impact of individual food components on the formation of immunity and educating them on the subject of safe supplementation. These activities can be undertaken by nutritionists, specialists, or doctors and directed to people from their earliest years, which will allow them to create healthy eating habits and make rational choices in later life to ensure and maintain full health.

## 5. Strengths and Limitations

The work raises the extremely important issue of human immunity. In the pandemic situation of recent times, this is a very important aspect. The topic of the work deals with nutritional issues, the observance of which can greatly strengthen the human immune system and thus prevent excessive infections, taking into account both the SARS-CoV-2 virus and other epidemiological threats that may arise in the world in the future. During the pandemic period, the topic of immunity has become very popular in the world of scientific research, but it is necessary to continue to expand knowledge in this area and provide more new and reliable research in this field to increase public awareness and affect a better quality of life for future generations.

In this study, a limitation was the problem of reaching a larger group of respondents, especially the male gender. This problem may be because men are already tired of the topic of coronavirus, so seeing a topic related to it, they were reluctant to participate in the survey. In addition, they are a minority in the Polish population and this is a typical situation in most scientific studies. Despite this limitation, this work joins the collection of works providing reliable information that can help future generations. Another issue limiting the work is the chosen methodology of the online form, which is why the authors introduced several facilities and safeguards, as described earlier, which helped minimize bias.

Finally, it is worth emphasizing that the topic of building immunity is a multifaceted thread and does not depend only on the adopted nutritional model, but it is important to remember that it is one of the important bricks that affect the lifelong process of taking and rebuilding immunity. Hence, the data presented were treated as a single thread without taking into account possible confounding variables.

## 6. Conclusions

The study’s hypothesis was supported. Respondents characterized by a normal dietary pattern and taking supplements commonly recognized as immune “boosters” were more likely to survive a mild episode of COVID-19. Moreover, it was shown that the pandemic in most of the respondents did not significantly affect their dietary strategy. It is reasonable to conclude that the dietary patterns adopted by the subjects may be a common way to prevent SARS-CoV-2 infections and their possible complications.

Future research should focus on the feasibility of introducing immune-supporting ingredients into the diet and on identifying such ingredients to build immunity against infectious diseases.

## Figures and Tables

**Table 1 life-12-01976-t001:** Characteristics of the study group (*n* = 304).

**Age**	
18–35	189 (62%)
36–55	64 (21%)
56–81	51 (17%)
**Place of residence**	
Village City of up to 50,000 residents City of 50,000 to 150,000 residents. A city with a population of more than 150,000	84 (28%) 51 (17%) 74 (24%) 95 (31%)
**Education**	
Basic	12 (4%)
Professional	32 (11%)
Medium	96 (32%)
Higher	164 (54%)
**Type of work performed**	
Pensioner	45 (15%)
Non-working person	9 (3%)
Pupil/student	65 (21%)
Mental work	117 (38%)
Manual labor	68 (22%)
**Occupation represented**	
Medical	50 (16%)
Non-medical	176 (59%)
Not applicable	75 (25%)

**Table 2 life-12-01976-t002:** Regularity of respondents’ consumption of preparations containing specific substances.

Variable (*N* = 304)	Regularity of Intake of Preparations Containing Certain Substances						
	Regularly	Irregularly	Don’t Use	Total (%)	T	r	*p*-Value *
**Vitamin A**	46 (15.1%)	34 (11.2%)	224 (73.7%)	304 (100%)	2.677	0.001	*p* = 0.11277
**Vitamin C**	102 (33.6%)	77 (25.3%)	125 (41.1%)		1.867	0.021	*p* = 0.20002
**Vitamin D**	140 (46.1%)	79 (26%)	85 (28%)		3.860	0.045	*p* = 0.81863
**Vitamin E**	36 (11.8%)	36 (11.8%)	232 (76.3%)		0.579	0.002	*p* = 0.82628
**Multivitamin**	50 (16.4%)	61 (20.1%)	193 (63.5%)		12.387	0.743	*p* = 0.04272 *
**Selenium**	25 (8.2%)	17 (56%)	262 (86.2%)		1.387	0.001	*p* = 0.73983
**Zinc**	46 (15.1%)	31 (10.5%)	226 (74.3%)	303 (99.67%)	2.370	0.001	*p* = 0.48279
**Probiotics**	15 (4.9%)	45 (14.8%)	244 (80.3%)	304 (100%)	1.654	0.031	*p* = 0.71194
**Acids** **omega-3**	69 (22.7%)	70 (23%)	165 (54.3%)		2.657	0.005	*p* = 0.86683

* Statistical significance level was determined for the educational level of study participants.

**Table 3 life-12-01976-t003:** Incidence of COVID-19 disease according to regular intake of preparations containing specific substances.

Variable	Incidence of Mild or Scanty COVID-19						
	Not (Severe)	Unspecified	Yes (Mild/Scanty)	Total (%)	T	r	*p*-Value
**Vitamin A**	10 (21.74%)	9 (19.57%)	27 (58.70%)	46 (100%)	1.768	0.031	*p* = 0.87351
**Vitamin C**	24 (23.53%)	17 (16.67%)	61 (59.80%)	102 (100%)	11.374	0.611	*p* = 0.04603 *
**Vitamin D**	32 (22.86%)	27 (19.29%)	81 (57.86%)	140 (100%)	2.987	0.002	*p* = 0.35587
**Vitamin E**	9 (25.00%)	8 (22.22%)	19 (52.78%)	36 (100%)	2.674	0.001	*p* = 0.60783
**Multivitamin**	10 (20.00%)	6 (12.00%)	34 (68.00%)	50 (100%)	13.456	0.711	*p* = 0.02191 *
**Selenium**	5 (20.00%)	5 (20.00%)	15 (60.00%)	25 (100%)	1.785	0.011	*p* = 0.89731
**Zinc**	9 (19.57%)	8 (17.39%)	29 (63.04%)	46 (100%)	0.756	0.056	*p* = 0.16594
**Probiotics**	9 (60.00%)	3 (20.00%)	3 (20.00%)	15 (100%)	0.324	0.001	*p* = 0.70255
**Omega-3 acids**	18 (26.09%)	15 (21.74%)	36 (52.17%)	69 (100%)	0.864	0.006	*p* = 0.67602

* Statistical significance level was determined for the education of study participants.

**Table 4 life-12-01976-t004:** Frequency of consumption of specific products by respondents.

Frequency of Consumption of Specific Products						
Product	Several Times Daily	Once Daily	Several Times a Week	Several Times a Month	Occasionally	Not I Consume
**Red meat,** **poultry**	14 (4.6%)	56 (18.4%)	139 (45.7%)	64 (21.1%)	22 (7.2%)	9 (3%)
**Offal (liver)**	1 (0.3%)	1 (0.3%)	5 (1.6%)	42 (13.8%)	112 (36.8%)	143 (47%)
**Fish**	0 (0%)	4 (1.3%)	30 (9.9%)	152 (50%)	98 (32.2%)	20 (6.6%)
**Seafood (oysters, crabs,** **shrimps)**	0 (0%)	1 (0.3%)	6 (2%)	28 (9.2%)	85 (28%)	184 (60.5%)
**Eggs**	4 (1.3%)	22 (7.2%)	173 (56.9%)	85 (28%)	18 (5.9%)	2 (0.7%)
**Milk and products** **milky**	49 (16.1%)	65 (21.4%)	120 (39.5%)	44 (14.5%)	17 (5.6%)	9 (3%)
**Milk products** **fermented** **(buttermilk, kefir)**	4 (1.3%)	17 (5.6%)	83 (27.3%)	82 (27%)	52 (17.1%)	66 (21.7%)
**Cheese** **rennet** **ripened (yellow cheese)**	9 (3%)	29 (9.5%)	118 (38.8%)	91 (29.9%)	42 (13.8%)	15 (4.9%)
**Curd cheeses**	4 (1.3%)	11 (3.6%)	108 (35.5%)	107 (35.2%)	57 (18.8%)	17 (5.6%)
**Products** **whole grains** **(bread, groats, rice, poppy seeds)**	60 (19.7%)	65 (21.4%)	111 (36.5%)	45 (14.8%)	20 (6.6%)	3 (1%)
**Pickles (cabbage, pickled cucumbers)**	3 (1%)	10 (3.3%)	68 (22.4%)	149 (49%)	59 (19.4%)	15 (4.9%)
**Citrus fruits** **(Oranges,** **lemons, kiwis)**	12 (3.9%)	23 (7.6%)	94 (30.9%)	121 (39.8%)	47 (15.5%)	7 (2.3%)
**Vegetables**	118 (38.8%)	63 (20.7%)	88 (28.9%)	24 (7.9%)	10 (3.3%)	1 (0.3%)
**Garlic, onion**	16 (5.3%)	39 (12.8%)	156 (51.3%)	48 (15.8%)	26 (8.6%)	19 (6.3%)
**Nuts** **(Brazilian,** **Italian, pecan,** **cashews)**	2 (0.7%)	11 (3.6%)	74 (24.3%)	79 (26%)	104 (34.2%)	34 (11.2%)
**Seeds, seeds** **(Sunflower, pumpkin,** **almonds)**	4 (1.3%)	8 (2.6%)	58 (19.1%)	82 (27%)	107 (35.2%)	45 (14.8%)
**Plant seeds** **Legumes** **(beans, peas,** **chickpea)**	0 (0%)	7 (2.3%)	32 (10.5%)	109 (35.9%)	127 (41.8%)	29 (9.5%)
**Vegetable fats**	20 (6.6%)	54 (17.8%)	121 (39.8%)	56 (18.4%)	43 (14.1%)	10 (3.3%)
**Animal fats**	26 (8.6%)	56 (18.4%)	85 (28%)	49 (16.1%)	59 (19.4%)	29 (9.5%)
**Herbs**	47 (15.5%)	45 (14.8%)	85 (28%)	40 (13.2%)	59 (19.4%)	28 (9.2%)

## Data Availability

Not applicable.

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
