# Peer review of "Proper Dietary and Supplementation Patterns as a COVID-19 Protective Factor (Cross-Sectional Study-Silesia, Poland)"

_life, 2022, doi:10.3390/life12121976_

Round 1

Reviewer 1 Report (Previous Reviewer 1)

The authors have revised the work nicely.

Author Response

Dear Reviewer,

we sincerely thank you.

Reviewer 2 Report (New Reviewer)

My comments are attached herewith

Author Response

Dear Reviewer,

Thank you for your work and time in reviewing the manuscript. The suggestions you indicated will be taken into account in full, and the manuscript, corrected in these respects, will be sent for further processing.

This manuscript is a resubmission of an earlier submission. The following is a list of the peer review reports and author responses from that submission.

Round 1

Reviewer 1 Report

Karolina Krupa-Kotara and colleagues have provided a survey-based research outcome for dietary supplement implementation for long COVID-19 issues. I recommend the following points to be addressed before any decision is made.

1. English needs to be improved

2. There are punctuation errors

3. Define all the abbreviations while the first instance of usage

4. Material and Method part there is no clarity of thought. Simplify it.

5. Why age is written like 39.04±23.59??

6. the severe acute 42 respiratory syndrome coronavirus 2 (SARS-CoV-2) virus?

7. Introduction is not up to the mark

8. Figures are of poor quality with very short captions!

9. In section 2: How bias was avoided? What is the guarantee that it is filled properly?

10. Define inclusion/Exclusion criteria properly

11. Study limitations should be discussed properly

Author Response

Dear Reviewer,

Thank you for your input, your time and the work you put into helping me prepare a better version of the manuscript. Thank you for the lapses you caught and any errors that may have contributed to a misunderstanding of the idea of the article. Any changes made to the manuscript are marked in red and address the suggestions indicated in your review, namely:
- the linguistic layer of the work and any punctuation errors were corrected;
- the title and purpose of the study were corrected to relate directly to the content;
- the abbreviations used have been corrected, i.e. before each abbreviation there is an explanation of it;
- the methodology has been rebuilt, new information has been added, especially in terms of eligibility criteria, validation of the questionnaire and additional data analysis;
- the recording of respondents' ages and reporting of results in the text was improved (test scores, correlation coefficients and probability levels were added);
- all figures have been removed;
- explained (methodology and limitations) how bias in the study was avoided;
- expanded the section on the study's main strengths and limitations;
- the conclusions were rewritten to respond to the hypothesis;
- the overall understanding of the study was improved, as previously the idea of surveying people after COVID-19 and seeing how their dietary and supplementation patterns affected the course of infection was misunderstood.

Thank you!

Reviewer 2 Report

The subject is not interesting and methodology is weak. I could not find a rational for this study. Maybe changing the analyses improve results.

Author Response

We extremely regret that the reviewer did not find a bit of time to describe more broadly his objections to the manuscript. We were forced to rely on the reviews of the other reviewers, which had a significant effect on strengthening the paper and improving its argumentation. We hope that the current form of reporting the results and presenting our theses is sufficient.

Below I present the responses that were sent to reviewer 1 and 3, included here are all the changes made:

Dear Reviewer,

Thank you for your input, your time and the work you put into helping me prepare a better version of the manuscript. Thank you for the lapses you caught and any errors that may have contributed to a misunderstanding of the idea of the article. Any changes made to the manuscript are marked in red and address the suggestions indicated in your review, namely:
- the linguistic layer of the work and any punctuation errors were corrected;
- the title and purpose of the study were corrected to relate directly to the content;
- the abbreviations used have been corrected, i.e. before each abbreviation there is an explanation of it;
- the methodology has been rebuilt, new information has been added, especially in terms of eligibility criteria, validation of the questionnaire and additional data analysis;
- the recording of respondents' ages and reporting of results in the text was improved (test scores, correlation coefficients and probability levels were added);
- all figures have been removed;
- explained (methodology and limitations) how bias in the study was avoided;
- expanded the section on the study's main strengths and limitations;
- the conclusions were rewritten to respond to the hypothesis;
- the overall understanding of the study was improved, as previously the idea of surveying people after COVID-19 and seeing how their dietary and supplementation patterns affected the course of infection was misunderstood.

Reviewer 3 Report

“Comments to the Author”

1. The present article explores the patterns of diet and nutritional supplementation of adults during the COVID-19 pandemic, which are aligned with the scopes and expectations of Life’s journal and its readers.

2. However, this manuscript has a significant limitation regarding the clearly defined study’s objective which led to purposeless data handling, discussion, and conclusion.

 Major remark

1.      What is the study’s objective?

From the article titled “Dietary and supplementation management strategy for building immunity in the pandemic and post-pandemic period (COVID-19)”, the abstract “ The purpose of this study is to evaluate dietary 25 management strategies and supplementation in building immunity during the COVID-19 pandemic 26 in the adult population.”, and at the end of the introduction (line 109-112) “ It is hypothesized that proper nutrition and supplementation of immunostimulants 109 may act as a preventive measure in the aspect of SARS-CoV-2 virus infection. Therefore, 110 the purpose of this study is to evaluate dietary management and supplementation 111 strategies in building immunity during the COVID-19 pandemic in the adult population.”

These sentences implied that the study intended to analyze participants’ dietary/supplementation changes during the pandemic that may have impacts on host immunity.  

However, the data gathering and handling processes mostly described the dietary/supplementary patterns of participants during the survey. The analytic processes also did not focus on the immunity outcomes. It was clear that the independent and dependent variables were not defined in the study.

To align with the stated objective, data collection may need to include people who do and do not change their diet/supplementation patterns, together with each category of diet and supplementation component. These variables were then tested against the immunity outcomes whether they had or had not gotten COVID-19. This can also be done with other lifestyle variables such as sleep, physical activities, stress levels, etc.

The confounding factors, that impact host immunities, were needed to be addressed and controlled, including age ranges, vaccination history,  comorbidities, etc.

2.      Another option might be the changes of title and study objective to comply with the data handling and analysis.

Minor remarks

3.      Without a clear data analytic direction, few outputs contradict the main authors’ argument, such as the people who regularly take multivitamins ( 50, p 0.04272, table 2) were associated with a high occurrence of COVID-19 ( 34, p 0.0219, table 3).

4.      The inaccuracies of terms were found all over the article. For instance, in line 254-255, “The largest percentage of respondents with COVID-19 disease regularly took vitamin 254 C-containing preparations/drugs/supplements (n=61; 59.80%).” The statement implies the highest number of participants that took vitamin C and got COVID-19 (61), not the percentage (59.80%). The highest percentage (68%) was the one in the multivitamin group.

5.      Another instance of term inaccuracy was in the introduction part and figure 1 regarding the tasks of the immune system. Clear references were needed before composing the figure. Figure 1 represented only part of immune functions. The authors may consider referring to the classical immune categories, such as mucosal/systemic immunity, or intrinsic/innate/acquired immunities. The concept of “increasing the body’s immunity (line 62)”, and many places throughout the article, was already obsolete. You may need to be careful with these terms.

6.      When we introduce abbreviations in the article, we should use them following the full term. For example, line 42, “The COVID-19 (coronavirus disease 2019) pandemic…”, would be better with “ The coronavirus disease 2019, or COVID-19, pandemic….)

7.      Language editing could help to improve the manuscript's arguments, clarity, and accuracy. I suggest rewriting it.

Author Response

(The authors gave the same response as above.)

Round 2

Reviewer 1 Report

The authors have revised the manuscript as per my suggestions.

Reviewer 3 Report

Thank you for the reviewed manuscript. The study’s objective is modified and moved toward a proper direction. However, the whole article is expected to be adapted to support this new goal, including the introduction, data analysis, table, discussion, and conclusion. This work should also be done through the collective efforts of all co-authors. The authors’ responses are also expected as point-to-point rectifications. The preparation quality of the article is still poor. For instance, the study’s objective in the abstract remains unchanged despite the major modification in the main manuscript. I suggest rewriting the whole article.